# Floating ZnO QDs-Modified TiO_2_/LLDPE Hybrid Polymer Film for the Effective Photodegradation of Tetracycline under Fluorescent Light Irradiation: Synthesis and Characterisation

**DOI:** 10.3390/molecules26092509

**Published:** 2021-04-25

**Authors:** Anwar Iqbal, Usman Saidu, Farook Adam, Srimala Sreekantan, Noorfatimah Yahaya, Mohammad Norazmi Ahmad, Rajabathar Jothi Ramalingam, Lee D. Wilson

**Affiliations:** 1School of Chemical Sciences, Universiti Sains Malaysia, Gelugor, Penang 11800, Malaysia; usmaniyya2000@gmail.com (U.S.); farook@usm.my (F.A.); 2School of Materials & Mineral Resources Engineering, Universiti Sains Malaysia, Nibong Tebal, Penang 14300, Malaysia; srimala@usm.my; 3Integrative Medicine Cluster, Advanced Medical and Dental Institute, Universiti Sains Malaysia, Bertam, Kepala Batas, Penang 13200, Malaysia; noorfatimah@usm.my; 4Experimental and Theoretical Research Lab, Department of Chemistry, Kulliyyah of Science, Kuantan, Pahang 25200, Malaysia; mnorazmi@iium.edu.my; 5Surfactant Research Chair, Chemistry Department, College of Science, King Saud University, P.O. Box 2455, Riyadh 11451, Saudi Arabia; 6Department of Chemistry, University of Saskatchewan, 110 Science Place, Room 165 Thorvaldson Building, Saskatoon, SK S7N 5C9, Canada

**Keywords:** LLDPE, zinc oxide, titanium dioxide, quantum dots, tetracycline, photodegradation

## Abstract

In this work, mesoporous TiO_2_-modified ZnO quantum dots (QDs) were immobilised on a linear low-density polyethylene (LLDPE) polymer using a solution casting method for the photodegradation of tetracycline (TC) antibiotics under fluorescent light irradiation. Various spectroscopic and microscopic techniques were used to investigate the physicochemical properties of the floating hybrid polymer film catalyst (8%-ZT@LLDPE). The highest removal (89.5%) of TC (40 mg/L) was achieved within 90 min at pH 9 due to enhanced water uptake by the LDDPE film and the surface roughness of the hybrid film. The formation of heterojunctions increased the separation of photogenerated electron-hole pairs. The QDs size-dependent quantum confinement effect leads to the displacement of the conduction band potential of ZnO QDs to more negative energy values than TiO_2_. The displacement generates more reactive species with higher oxidation ability. The highly stable film photocatalyst can be separated easily and can be repeatedly used up to 8 cycles without significant loss in the photocatalytic ability. The scavenging test indicates that the main species responsible for the photodegradation was O_2_^●^^−^.

## 1. Introduction

The efficiency of conventional wastewater treatment plants in removing antibiotics lies in the range of 12% to 80%. The annual consumption of antibiotics in the world is estimated to be around 200,000 tons [1]. Much of the consumed antibiotics are incompletely metabolised in the body, which results in ca. 30% to 90% of the remaining antibiotic entering the ecosystem either as the parent compound or as by-product metabolites [2]. Current conventional wastewater treatment plants are ineffective in the removal of antibiotics at trace levels [3]. Hence, there is an urgent need to develop wastewater treatment systems to tackle this global water security issue.

The utilisation of direct solar light or solar simulation by photocatalysis can serve as an alternative to promote the degradation of antibiotics in the aquatic environment to harmless by-products [1]. Among the semiconductor photocatalysts, TiO_2_ is the most commonly used photocatalyst due to its low cost, high activity, excellent photostability, chemical inertness, and nontoxicity [4]. However, the high electron-hole recombination rate makes TiO_2_ a relatively inefficient photocatalyst. Heterojunction formation between the TiO_2_ and semiconductor quantum dots (QDs) has proven to be one of the most effective approaches to overcome this drawback. QDs’ modified titania catalyst promotes a charge separation and minimises or inhibits electron/hole recombination for improved photocatalytic activity [5,6,7]. The semiconductor QDs act as sensitisers or separation centres when deposited onto the surface of TiO_2_. Fabrication of bulk semiconductor with QDs may also maximise the specific surface area of the photocatalyst due to their reduced particle sizes, and the migration distance required for electron/hole pairs to reach the surface is minimised [5,6]. Many semiconductor QDs have been successfully deposited on TiO_2_ that all showed improved photocatalytic activity when compared with pristine TiO_2_. For instance, WO_3-x_ QDs [7], SnO_2_ QDs [8], CdSe QDs [6], and PbS QDs [9] have been reported to exhibit enhanced photocatalytic properties. However, the photocatalysts’ main drawback is the difficult separation from the water system for their reusability. This relates to the common use of a powdered form of the photocatalysts, where they function as suspended colloidal particles in wastewater to degrade the organic contaminants. Other drawbacks include low light utilisation efficiency of the suspended photocatalyst; aggregation of catalytic particles, especially at higher concentrations, and potential human health problems associated with the fate and transport of dispersed powder and its mobility [10,11]. These drawbacks limit their application in large-scale commercial production. An efficient alternative to address this issue relates to the immobilisation of the nanoparticles onto various substrates, such as glass fibres, clay beads, ceramics, and polymers [12,13,14,15]. Moreover, most of the supports are either expensive, contain potential impurities, or display reduced durability and flexibility [16]. Polymer supports have proved to be among the best candidates for immobilising photocatalyst nanoparticles due to their numerous advantages: low cost, ready accessibility, chemical inertness, and mechanical stability with high durability. In addition, polymer supports have relatively low density, allowing for the design of floating photocatalysts, especially in aqueous media [11,17,18,19].

Linear low-density polyethylene (LLDPE) exhibits excellent chemical, mechanical, and thermal stability [17]. It can also be used to fabricate thinner films with better environmental stress cracking resistance. In addition, its low-density property offer the option to produce floatable-type photocatalyst systems. Some of the most important advantages of such photocatalysts relate to their maximum utilisation of solar energy, where they can be directly applied into the contaminated wastewater treatment efficiently under non-stirred conditions to achieve cost reduction for photocatalysis [20,21].

In this work, floating ZnO QDs-modified TiO_2_ was supported on an LLDPE hybrid film prepared by a facile and eco-friendly solution-casting method. The photocatalytic activity and proposed mechanisms of the prepared hybrid film were evaluated for the photodegradation of tetracycline (TC) in an aqueous medium under fluorescent light irradiation. The effect of various parameters and a kinetic study of the photodegradation degradation rate for TC was investigated using the modified Langmuir–Hinshelwood method.

## 2. Results and Discussion

### 2.1. Characterisation

#### 2.1.1. X-ray Diffraction (XRD) Analysis

The XRD profiles of ZnO QDs, ZT, bare LLDPE, and the 8%-ZT@LLDPE are presented in Figure 1a–d, respectively. The XRD diffraction planes observed in the diffractogram of ZnO QDs (Figure 1a) are indexed to the hexagonal Wurtzite phase (JCPDS 01-071-6424). The size of the ZnO QDs (4.4 nm) was determined using the Scherrer equation, according to Equation (1).

(1)
$$D=\frac{k\lambda }{\beta cos\theta }$$

*D* is the crystal size, *λ* is the wavelength equal to 0.154 nm, *k* is a constant taken as 0.94, *β* is the full width at half-maximum intensity (FWHM) of the XRD band, and *θ* is the Bragg angle of XRD band. The diffraction bands at the following 2*θ*-values of 25.4°, 37.9°, 48.0°, 54.1°, 55.1°, and 62.9° (Figure 1b) correspond to the (101), (004), (200), (105), (211), and (204) planes of anatase TiO_2_ phase (JCPDS 00-021-1272) [22]. The average crystallite size of the ZT determined from equation (1) was 9.33 nm. The diffraction peaks related to the ZnO QDs were not observed in the XRD profile of ZT. Several reasons can account for the absence of the peaks: (1) The concentration of the ZnO QDs is low, where Han et al. [23] reported a similar observation upon incorporation of CdSe QDs into the framework of ZnO flowers. (2) The effect may relate to the level of the homogeneity of the mixing of Ti and Zn components in ZT [24]. This means that anatase TiO_2_ and ZnO QDs did not exist as a separate phase. (3) Incorporation of ZnO QDs into the lattice of TiO_2_ led to a change in the pH of the media, where the high calcination pH may have induced a strain on the crystal framework of the ZnO QDs. The induced strains can distort the crystal framework, hence favouring an amorphous product. In turn, the XRD bands related to ZnO crystal structure were not detected in XRD profile. From the diffractogram of the bare LLDPE film (Figure 1c), XRD bands at 2*θ* values of 21.3°, 23.6°, and 36.0° were detected that correspond to (110), (200), and (020) planes of an orthorhombic polyethylene crystal phase, respectively [25]. The presence of ZT within the matrix of LLDPE is evident with the emergence of an anatase peak (101) at a 2*θ* = 25.4°. The diffraction peaks related to the LLDPE and anatase TiO_2_ were observed to be broadened and attenuated. The broadening of the XRD lines and their weaker intensity indicates that LLDPE became less crystalline after ZT underwent immobilisation.

#### 2.1.2. Field Emission Scanning Electron Microscopy (FESEM) Analysis

The FESEM analysis was carried out to understand the morphology of the ZT and its influence on the LLDPE film’s morphology. The FESEM image of ZT (Figure 2a) shows the presence of pores on the surface of ZT. The average pore diameter was measured to be ~ 1.17 µm. The pores were formed due to the use of starch as a template in the synthesis. Hydrogen bonds in the starch granules are weakened to allow more sorption of water molecules upon swelling of the starch granules. Based on the concept of the partial charge model, the hydrolysis of a titanium cation that forms a stable [Ti(OH)(OH_2_)_5_]^3+^. As the pH increases, deprotonation of [Ti(OH)(OH_2_)_5_]^3+^ forms [Ti(OH)_2_(OH_2_)_5_]^2+^. Condensation takes places when [Ti(OH)_2_(OH_2_)_5_]^2+^ is further deprotonated to [TiO(OH)(OH_2_)_4_]^+^, which undergoes intramolecular deoxolation to [TiO(OH)_3_(OH_2_)_3_]^+^ [26,27]. As the condensation takes place, the species is drawn to the starch granules by van der Waals attraction. A three-dimensional oxide framework incorporating the ZnO QDs forms around the swollen starch granules. The starch granules are removed during calcination, leaving behind a TiO_2_/ZnO QDs embedded within a porous nanocomposite oxide framework.

From the FESEM image of bare LLDPE film (Figure 2b), the surface of the film appeared to be virtually smooth. The molten LLDPE was observed to have higher viscosity in the presence of ZT. As it solidified, the surface of the film became uneven (Figure 2c). The bright spots on the SEM image (shown using green arrows) were probably related to ZnO QDs [28]. A cross-section of the 8%-ZT@LLDPE (Figure 2d) indicates that the ZT nanocomposite was positioned on the film and free from any voids or microcracks. The average thickness of the film was measured as 120 µm.

#### 2.1.3. Field Emission Scanning Electron Microscope (FESEM) Analysis

The TEM images of the ZnO QDs and ZT are shown in Figure 3. The histograms are given as an insert therein. The particles were irregular and spherically shaped with evidence of agglomeration. The average size of ZT was 12.5 ± 2.05 nm, whereas the average size of the ZnO QDs were 4.12 ± 0.23 nm.

#### 2.1.4. Atomic Force Microscope (AFM) Analysis

Surface roughness is a vital aspect to be considered to afford enhanced photocatalytic activity of a film-based photocatalyst. A higher degree of surface roughness would improve the hydrophilicity of the films to enhance the interaction of the film with oxygen, water, and pollutants for the reaction to occur [29,30]. Figure 4 shows the three-dimensional view of the AFM images (5 µm × 5 µm) for the bare LLDPE film and the 8%-ZT@LLDPE. The surface mean roughness (Ra) was 103 nm, and 273 nm for both the bare LLDPE and 8%-ZT@LLDPE, in agreement with trends reported from other studies. Hir et al. [27] reported the mean surface roughness of a polyethersulfone-TiO_2_ (PT) film increased from 2.19 nm to 4.46 nm for neat PES to PES immobilised with 13 wt.% TiO_2_ (PT-13). Similarly, Mohamed et al. [30] reported an increase in the mean surface roughness of regenerated cellulose/N-doped TiO_2_ (RC/TiO_2_) membrane from 29.53 nm to 79.11 nm for bare RC and RC/TiO_2_-0.7, respectively.

#### 2.1.5. Contact Angle Results

A liquid in contact with a solid will exhibit a contact angle (*θ*). The *θ*-value is determined by the interaction of the three interfaces (solid, liquid, and gas). A *θ*-value smaller than 90° indicates an attraction, where the liquid will spread to some extent over the solid surface. A *θ*-value above 90° indicates less attraction, where the liquid will minimise its contact area with the solid surface and form a droplet. Figure 5 displayed variable *θ*-values for water onto bare LLDPE and 8%-ZT@LLDPE. The increase in the hydrophilicity was due to the decrease in the composite films’ crystallinity as the ZT was added. It has been reported that a lower degree of crystallinity allows a polymer to absorb more water [31], which is beneficial for the photocatalytic activity.

#### 2.1.6. UV–Visible Diffuse Reflectance Analysis

The absorption spectra of ZT, TiO_2_, ZnO QDs, bare LLDPE, and 8%-ZT@LLDPE are shown in Figure 6. The bare LLDPE shows a broad absorption spectrum. The sensitivity of 8%-ZT@LLDPE towards the UV light region was higher due to the presence of the ZT nanocomposite. The increase in UV light sensitivity is caused by ZnO and TiO_2,_ which are known to be good UV absorber materials. The band edge of ZnO QDs, TiO_2_, and 8%-ZT@LLDPE was determined to be 377–380 nm, which is in the UV–Vis region. 

The bandgap energies of the ZnO QDs and TiO_2_ were estimated using Tauc’s equation, shown in Equation (2).

(2)
$$\alpha h\nu =A{\left(h\nu -Eg\right)}^{n/2}$$

α is the absorption coefficient, h is Planck’s constant, ν is the light frequency, *A* is the proportionality constant, and *Eg* is the bandgap energy. The bandgap energy of the semiconductors was estimated from a Tauc plot of (αhν)^1/2^ versus hν (Figure 6b). The estimated bandgap was 3.10 eV and 3.50 eV for TiO_2_ and ZnO QDs, respectively.

### 2.2. Photocatalytic Activity of the LLDPE Hybrid Films

#### 2.2.1. Effect of Various Photocatalysts Immobilised on LLDPE Film

The effect of various nanocomposites immobilised onto a LLDPE film for photodegradation of TC is presented in Figure 7a. Tetracycline was not photodegraded without any catalyst (blank) or in the presence of bare LLDPE film. The TiO_2_@LLDPE removed 42.7% of TC, whereas ZnO QDs@LLDPE removed 53.4% of TC. Commercial TiO_2_ P25 (Degussa) immobilised on the LLDPE (P25@LLDPE) removed 20.5% of the TC, which is lower compared to TiO_2_@LLDPE. The presence of pores on the TiO_2_ surface resulted in attenuation of the recombination rate of photogenerated e-/h+ pairs, hence increasing the photocatalytic activity.

The highest removal (85.4%) of TC was obtained in the presence of 8%-ZT@LLDPE. Previously, it has been discussed that ZT distorted the crystalline framework of the LLDPE, and increased the water uptake and surface roughness. In addition, such enhancement was also attributed to the formation of a heterojunction between ZnO QDs and TiO_2_, favouring the separation of photogenerated e^−^/h^+^ pairs. The structural defects of the Ti species and oxygen vacancies can also suppress the recombination rate of photogenerated e^−^/h^+^ pairs. The size-dependent quantum confinement effect displaces the conduction band potentials of ZnO QDs to more negative energy values than TiO_2_, leading to the generation of reactive species with higher oxidation ability.

The TC degradation rate plot was obtained using the modified Langmuir–Hinshelwood equation (cf. Figure 7b), whereas the calculated rate constants and the corresponding correlation coefficients (*R*^2^) for the films are presented in Table 1. The 8%-ZT@LLDPE has the highest rate constant (0.01077 min^−1^). The progressive degradation of TC was confirmed by the steady decrease in the peak intensity at 358 nm, as illustrated by the UV–vis spectra in Figure 7c.

#### 2.2.2. Effect of the Initial pH of the Solution

The efficiency of the 8%-ZT@LLDPE in the photodegradation of TC at different pH values that range from pH 3 to pH 11 is shown in Figure 7d. After 180 min of irradiation, a variable TC photodegradation efficiency was observed, as follows: 40.76% (pH 3), 73.45% (pH 5.8), 85.43% (pH 9), and 56.42% (pH 11). The pK_a_ values of TC were reported to be 3.30, 7.68, and 9.69 [32]. Hence, TC will be positively charged in solution at pH ≤ 3, neutral in solution with pH from 3.3 to 7.7, and negatively charged in solution with pH > 7.7. The PZC of ZT was determined to be 7.1, as shown in Appendix A. Therefore, in the acidic medium, the positively charged surface of 8%-ZT@LLDPE will repel the positively charged TC. At pH 6, a reduced electrostatic attraction between the neutral TC molecule and the positively charged surface of the catalyst. Hence, an increase in the TC adsorption and photodegradation rate was observed. However, at pH 9, an opposite trend was observed. At alkaline pH, negatively charged TC species tend to attract reactive groups such as ^●^OH due to their high electronic density on the ring system, which would result in the enhancement of removal efficiency of TC [33,34]. However, the degradation rate of TC is inhibited when the pH > 10, since the ^●^OH will compete with TC to be adsorbed on the photocatalyst surface [35,36,37].

#### 2.2.3. Effect of Natural Organic Matter (NOM)

The concentration of NOM in surface waters typically varies from about 2 mg/L to more than 50 mg/L, depending on the origin [38], and about 90% of NOM is humic acid (HA) [39]. Thus, it was used as a NOM model in the present study. As shown in Figure 8a, the addition of 5 mg/L of HA did not significantly affect the TC degradation. However, when the HA concentration was increased to 10, 20, and 50 mg/L, the TC removal efficiency decreased to 76.91%, 66.07%, and 52.86%, respectively. The inhibitory effect of HA is ascribed to the competitive adsorption of the HA onto the active sites of the photocatalyst, and reduced light penetration through the solution [40]. In addition, HA can also act as a reactive species quencher, which decreases the availability of reactive species for TC degradation.

#### 2.2.4. Scavenging Test

To identify the types of photogenerated oxidative species and to propose a possible photocatalytic mechanism, scavenging tests were conducted. Triethanolamine (TEA), ascorbic acid (AA), and isopropanol (IPA) were used as scavengers for photogenerated holes (h^+^), superoxide radicals (O_2_^●^^−^), and hydroxyl radicals (^●^OH) [41], respectively. The experimental reaction conditions for scavenging were similar to the photodegradation experiment, except with the addition of 5 mM of scavengers prior to the light irradiation. As depicted in Figure 8b, the addition of AA resulted in the drastic decrease (68.61%) in TC photodegradation efficiency, which implied that the O_2_^●^^−^ radicals (which were trapped by AA in solution) play a significant role in the degradation of TC. When the IPA was added to the solution, about 26% of TC was degraded, whereas only about 20% decrease in TC degradation efficiency was obtained when TEA was added. Hence, it is concluded that the main photogenerated oxidative species responsible for the photodegradation of TC by 8%-ZT@LLDPE hybrid film is O_2_^●^^−^, followed by ^●^OH and h^+^.

#### 2.2.5. Mechanism of TC Degradation Using 8%-ZT@LLDPE

The possible photodegradation mechanism of TC degradation using 8%-ZT@LLDPE is shown in Figure 9. The potentials of the valence band (VB) and the conduction band (CB) edges of the catalyst were calculated using Mulliken electronegativity theory, using Equations (3) and (4):
(3)
$$E_{VB}=X-E^{e}+0.5Eg$$


(4)
$$E_{CB}=E_{VB}-Eg$$

*E_VB_* is the valence band edge potential, *E_CB_* is the conduction band potential, *Eg* is the bandgap of the semiconductor, 
$E^{e}$
 is the energy of free electrons on the hydrogen scale (~4.5 eV), and *X* is the electronegativity of the semiconductor, which is 5.62 eV and 5.81 eV for ZnO and TiO_2_, respectively. The calculated VB and CB potentials of ZnO QDs in 8%-ZT@LLDPE composite film are 2.87 and −0.46 eV, respectively; whereas the calculated VB and CB potentials for the TiO_2_ in 8%-ZT@LLDPE are 2.95 and −0.15 eV, respectively. Since the CB potential of the mesoporous TiO_2_ is less negative than the reduction potential of oxygen (E^0^(O_2_/ O_2_^●^^−^ = −0.33 eV/NHE), the TiO_2_ does not have sufficient reduction potential to generate O_2_^●^^−^. Based on the above experimental results and analysis, a direct Z-scheme mechanism in the photodegradation of TC using 8%-ZT@LLDPE is proposed. When irradiated with light, both electrons in the VB of ZnO QDs and mesoporous TiO_2_ will be excited to the CB, leaving behind photogenerated holes (h^+^). The photogenerated electrons (e^-^) in the CB of the mesoporous TiO_2_ has less redox potential compared to the ZnO QDs, hence they combine with the h^+^ in the VB of ZnO QDs. Meanwhile, the e^-^ in the CB of ZnO QDs have more negative potential than the standard redox potential of O_2_, E^0^(O_2_/ O_2_^●^^−^ = −0.33 eV vs. NHE). Thus, there is a reduction in the adsorbed oxygen to O_2_^●^^−^. The h^+^ that remained in the VB of the TiO_2_ has more positive potential than the standard redox potential of OH^−^, (E^0^(OH^−^/^●^OH) = 2.4 eV); hence, TiO_2_ can oxidise OH^−^ to produce ^●^OH. The h^+^ can also directly oxidise the adsorbed TC to intermediates and even CO_2_ and H_2_O. The directional migration of photogenerated electrons and holes in the 8%-ZT@LLDPE hybrid film prevented the recombination of charge carriers, leading to photocatalytic activity improvement. The proposed mechanism is further supported by the scavenging tests reported herein.

#### 2.2.6. Mineralisation Efficiency

In order to determine the mineralisation ability of the hybrid film, total organic carbon (TOC) analysis was conducted after photodegradation of TC under visible light irradiation. As shown in Figure 10a, the mineralisation efficiency of TC was observed to increase as the irradiation time increased. Moreover, the 8%-ZT@LLDPE hybrid film possessed higher mineralisation efficiency than Degussa P25@LLDPE. This is due to the effective formation of a heterojunction formation between the two semiconductors like ZnO QDs and TiO_2_, which favour the separation of photogenerated e^−^/h^+^ pairs and enhance the recombination rate for the photocatalytic process.

#### 2.2.7. Reusability

Reusability studies are of paramount importance in photocatalysis, as it provides insight on the photostability and robustness of the photocatalyst [42]. Reusability of 8%-ZT@LLDPE for the photodegradation of TC was tested under the optimum conditions. After each cycle, the film was removed and washed with deionised water and dried before being placed into another fresh solution of TC for another cycle. The process was repeated eight times, where the results are presented in Figure 10b. The prepared film with photocatalyst exhibited comparable degradation performance, with a slight reduction in performance (<5%) after 8 cycles of the photodegradation process. This remarkable stability provides support for the potential utility of this photocatalyst film for practical applications.

Table 2 presents the TC removal (%) and the required conditions of several reported immobilised photocatalyst systems in the literature. Based on the results in Table 2, it is noted that 8%-ZT@LLDPE requires a simple set up such as a common household fluorescent lamps and very mild conditions. By comparison, Shams et al. [43] recently reported the adsorption-based removal of TC (81.1%) from hospital wastewater using a vanadium oxide nano cuboid adsorbent material (cf. Table 8 in [43]). This level of physical (non-degradative) removal corresponds to a maximum adsorption capacity of 126.8 mg/g for TC under optimised experimental conditions: 20.5 °C, pH of 4.1, mixing time of 44.5 min, adsorbent dosage of 0.49 g/L, and initial concentration of TC = 30.7 mg/L. This is in contrast to the relatively efficient chemical transformation of TC (89.45%) reported in Table 2 by photocatalysis with 8%-ZT@LLDPE.

## 3. Materials and Methods

### 3.1. Materials

The LLDPE (LL1001 XV) was purchased from ExxonMobil Chemical (Singapore), with a melt flow index (MFI) of 1.0 g/10 min and a density of 0.918 g/cm^3^. Titanium (IV) tetraisopropoxide (TTIP, >98%), tetracycline (TC, >99%), and tetraethylorthosilicate (TEOS, 98%) were purchased from Acros Organics (Fair Lawn, NJ, USA). Zinc acetate dihydrate (99.8%), and soluble starch were from Fisher Scientific (Leicestershire, UK); whereas toluene (99.5%), ethanol (>99.7%), methanol, and potassium hydroxide (>95%) were purchased from QReC (Bangkok, Thailand), Sdn Bhd. All chemicals and reagents were of analytical grade and were used without further purification.

### 3.2. Synthesis of ZnO Quantum Dots (ZnO QDs)

The synthesis of ZnO QDs was carried out according to a reported method [51], with some modifications. A zinc acetate solution was prepared by dissolving 1.1 g of zinc acetate dihydrate in 100 mL of methanol under vigorous stirring. The pH of the solution was raised to pH 14 using KOH solution (1 M). The resulting solution was stirred continuously for 45 min using a magnetic stirrer. TEOS (0.25 mL) was added to the solution to stop the ZnO nucleation process, followed by the addition of 0.5 mL of distilled water to initiate the sol−gel reaction between the silica surface and the ZnO QDs. The ZnO QDs precipitate was recovered by centrifugation and washing thrice with methanol and distilled water to remove the unreacted precursors. The ZnO QDs powder was dried at 100 °C for 24 h.

### 3.3. Synthesis of ZnO QDs Modified TiO_2_

The ZT was synthesised according to the procedure reported by [52,53], with some modifications. A starch solution was prepared by dissolving 2 g of soluble starch in 200 mL of hot boiling water. Into the starch solution, 5.90 mL of TTIP was added and continuously stirred for 10 min at 65 °C. The pH of the solution was adjusted to pH 9 by adding ammonium hydroxide solution dropwise with constant stirring for 30 min. At this stage, 0.26 g of ZnO QDs was added to the TiO_2_ sol and stirred continuously for another 1 h at 85 °C. Finally, the mixture was centrifuged, and the product obtained was washed several times with ethanol and distilled water to remove any unreacted precursors. The product was dried at 100 °C for 12 h, and then calcined at 500 °C for 2 h with a heating rate of 3 °C min^−1^. The synthesised nanocomposite was labelled as ZT. Pristine TiO_2_ was synthesised without the ZnO QDs for comparison.

### 3.4. Synthesis of ZnO QDs Modified TiO_2_/LLDPE Floating Hybrid Polymer Film

The hybrid film was prepared by using a simple solution casting method, as reported by Saharudin et al. [31]. In a typical synthesis, 1.0 g LLDPE pellets were melted in 20 mL of toluene at 90 °C under continuous stirring for 20 min. An 8 wt.% (based on LLDPE mass) of ZT- toluene mixture was prepared by dispersing an appropriate amount of ZT in 10 mL of toluene, and sonicated for 5 min. The ZT- toluene mixture was added dropwise into the melted LLDPE and stirred for 5 min. Finally, the mixture was poured into a 90 mm diameter petri dish and oven-dried at 60 °C for 12 h. The TGA analysis indicated that the weight loss for the composite film was 8%. The value indicates that all the ZT added during the preparation was incorporated into the polymer matrix. Hence, the synthesised hybrid film was denoted as 8%-ZT@LLDPE. For comparison, bare LLDPE film, TiO_2_, and ZnO QDs immobilised on LLDPE were prepared in the same way and labelled as bare LLDPE, TiO_2_@LLDPE, and ZnO@LLDPE, respectively. The photographs of the films are shown in Appendix A.

### 3.5. Characterisation of the Hybrid Film Photocatalyst

The structural properties of the synthesised ZT and the hybrid film were investigated using various characterisation techniques. The X-ray diffraction (XRD) patterns were recorded in the 2*θ* range of 20°–70° with a scanning speed of 0.02° min^−1^ using an X-ray diffractometer (BRUKER AXS D8) with Cu–Kα radiation (λ = 0.15478 nm). The morphology was determined by high-resolution field emission scanning electron microscopy (FESEM), while the surface roughness was analysed using atomic force microscopy (AFM), Dimension EDGE, BRUKER model. The wettability or water contact angle (CA) of the hybrid films surface was measured using Data Physics OCA 20 Optical Contact Angle Goniometer, at ambient temperature, based on the Sessile Method. The probe liquid was distilled water, and a drop of 0.6 µL was used for every measurement. The lower values of contact angle (*θ*) represent a more hydrophilic surface. The average *θ*-value was averaged by measuring five different positions measured on a sample. The optical properties of the photocatalysts were measured using solid-state UV/vis diffuse reflectance spectroscopy (PerkinElmer Lambda 35 UV/VIS Spectrometer) method by scanning the reflectance of the samples in the range of 200 nm to 650 nm.

### 3.6. Photocatalytic Reactions

The photocatalytic activity of the hybrid film photocatalyst was investigated by monitoring the photodegradation of TC in aqueous solution using a homemade photoreactor equipped with two fluorescent lamps (24 W each). The light intensity was determined to be 104.4 W/m^2^, whereas the residual UV leakage was detected to be 0.40 W/m^2^. The measurement was done using Dual-Input Data Logging Radiometer (model PMA2100) equipped with visible, UVA, and UVB detectors. The schematic diagram of the photoreactor is shown in Appendix A. In a typical process, 5 × 5 cm of the hybrid film was immersed in a 100 mL of 40 ppm TC solution. Before light irradiation, the mixture was kept in the dark for 30 min to reach adsorption–desorption equilibrium. An aliquot was withdrawn at regular time intervals. The change in TC concentration was determined using UV–Vis spectrophotometer (UV2600, Shimadzu, Duisburg, Germany), and the removal efficiency of TC was calculated using the following Equation (5),

(5)
$$R=1-\frac{C_{t}}{C_{0}}$$

where 
$C_{0}$
 represents the initial concentration before fluorescent light irradiation, *C_t_* the concentration at a time interval, and *R* is the TC removal efficiency.

The mineralisation of TC was studied by monitoring the decrease in total organic carbon (TOC) content. The TOC content was measured with a Shimadzu 5000 TOC Analyzer equipped with an autosampler. The degree of mineralisation of TC was calculated using Equation (6):
(6)
$$T\left(\%\right)=\frac{TOC_{0}-TOC_{t}}{TOC_{0}}\times 100$$

where 
$TOC_{0}$
 represents the initial TOC concentration before oxidation reaction, and 
$TOC_{t}$
 is the concentration after oxidation reaction.

The reaction rate of photocatalytic degradation of TC was investigated using the modified Langmuir–Hinshelwood model (Equation (7)). The rate constant was obtained by plotting ln(*C*_0_/*C*) against time *t*, and the negative of the slope (*k*) represents the apparent reaction rate constant.

(7)
$$ln\;\left(\frac{C}{C_{0}}\right)=-kt\;$$

where *k* is the reaction rate (min^−1^), *t* is the irradiation time, and *C*_0_ and *C* are the initial and final concentrations of the antibiotic, respectively.

## 4. Conclusions

A floating TiO_2_-modified ZnO QDs/LLDPE hybrid polymer film photocatalyst (8%-ZT@LLDPE) was successfully prepared via a solution casting method. The AFM and water contact analyses indicate that the immobilisation resulted in an increase in the water uptake of LDDPE due to its greater surface roughness, which indirectly increased the TC adsorption properties. The formation of a heterojunction between ZnO QDs and the mesoporous TiO_2_ favours the separation of photogenerated electron-hole pairs. On the other hand, the QDs’ size-dependent quantum confinement effect leads to the displacement of the conduction band potentials of ZnO QDs to more negative energy values, as compared with mesoporous TiO_2_. The above-noted displacement leads to the generation of reactive species with higher oxidation ability. Under the optimum conditions (pH 9, 90 min), TC was successfully photodegraded up to 89.45%. The scavenging experiments reveal that superoxide radicals were the main active species responsible for the TC photodegradation. Moreover, the hybrid film retained its high photodegradation efficiency even after eight cycles. Hence, the prepared 8%-ZT@LLDPE has potential for the photocatalytic treatment of the toxic antibiotics present in industrial wastewater systems [47,54].

## Figures and Tables

**Figure 1 molecules-26-02509-f001:**
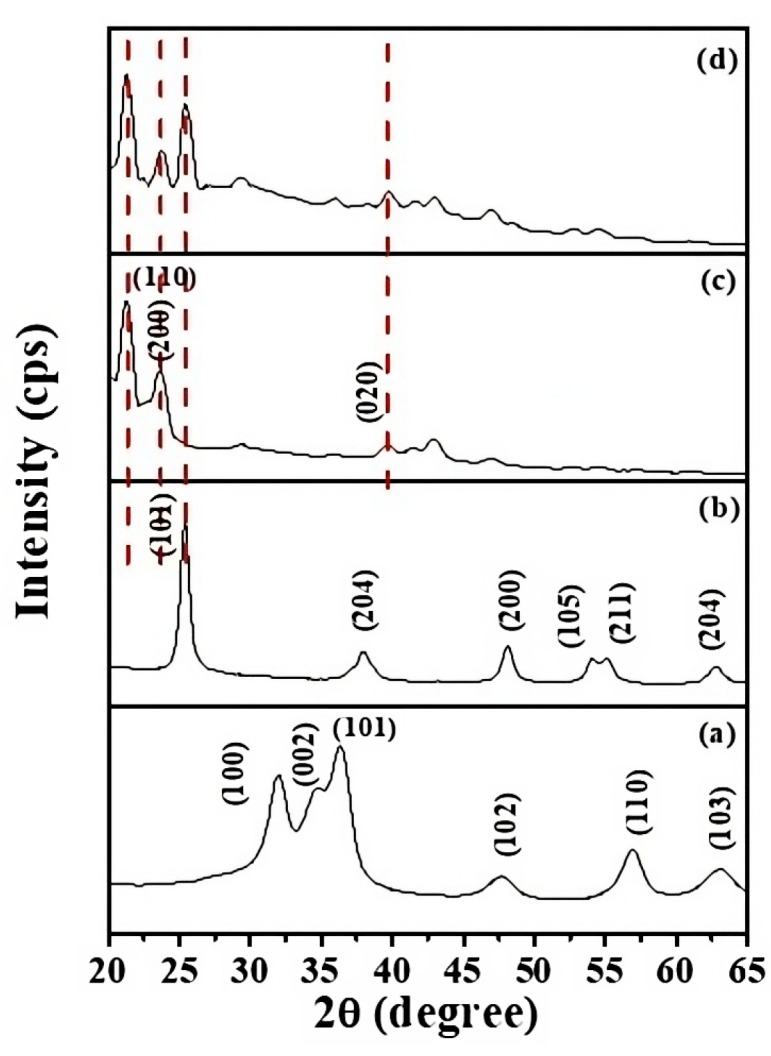
XRD diffractogram of materials: (**a**) ZnO QDs, (**b**) ZT, (**c**) bare LLDPE, and (**d**) 8%-ZT@LLDPE.

**Figure 2 molecules-26-02509-f002:**
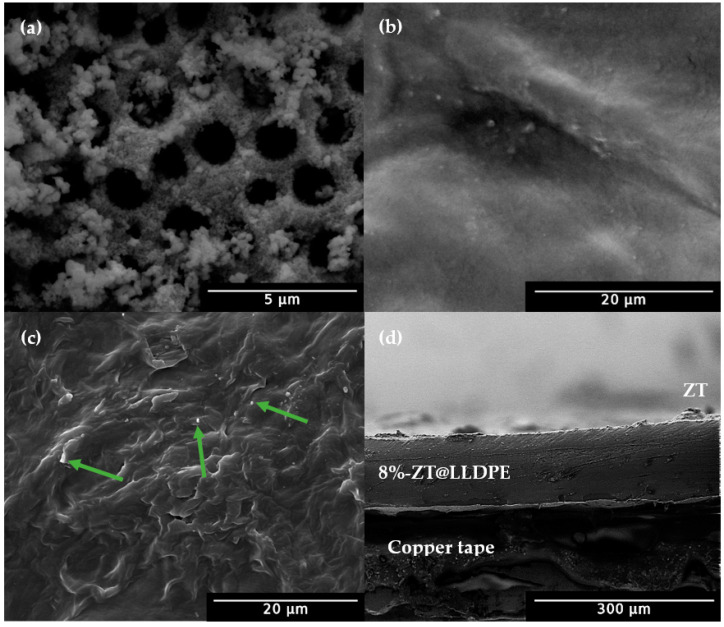
FESEM image of (**a**) ZT, (**b**) bare LLDPE, (**c**) 8%-ZT@LLDPE, and (**d**) cross-section of 8%-ZT@LLDPE. The green arrows indicate the ZnO QDs.

**Figure 3 molecules-26-02509-f003:**
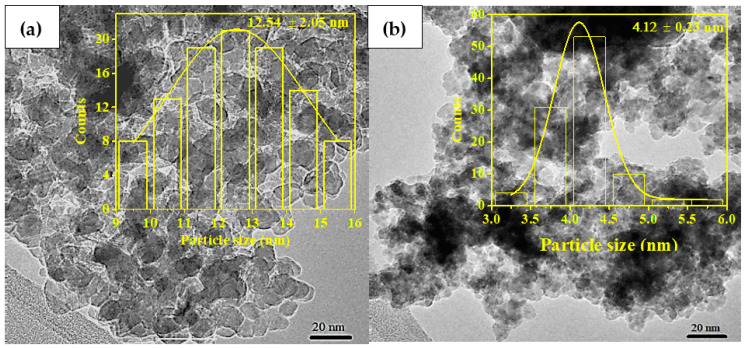
TEM image of (**a**) ZT and (**b**) ZnO QDs. The average particle size is shown in the insert.

**Figure 4 molecules-26-02509-f004:**
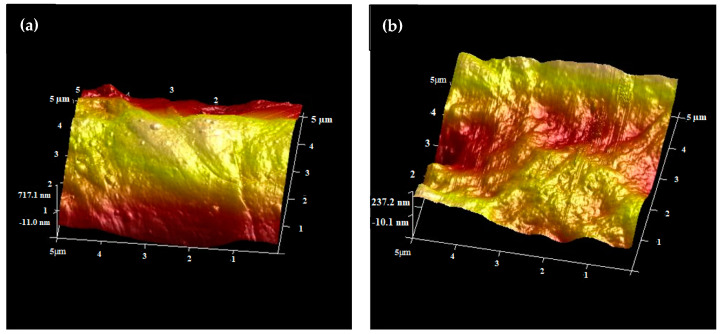
Three-dimensional AFM images of (**a**) bare LLDPE and (**b**) 8%-ZT@LLDPE.

**Figure 5 molecules-26-02509-f005:**
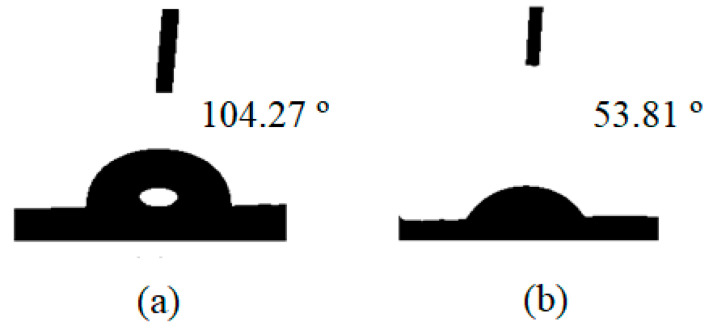
Water contact angle (*θ*) measurements: (**a**) bare LLDPE and (**b**) 8%-ZT@LLDPE.

**Figure 6 molecules-26-02509-f006:**
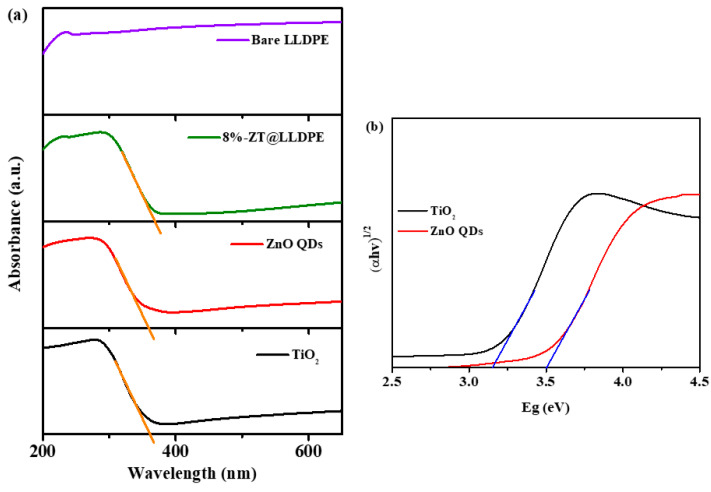
(**a**) Diffuse reflectance UV–Vis spectra of ZT, TiO_2_, ZnO QDs, bare LLDPE, and 8%-ZT@LLDPE; and (**b**) the Tauc plot of ZnO QDs and TiO_2_.

**Figure 7 molecules-26-02509-f007:**
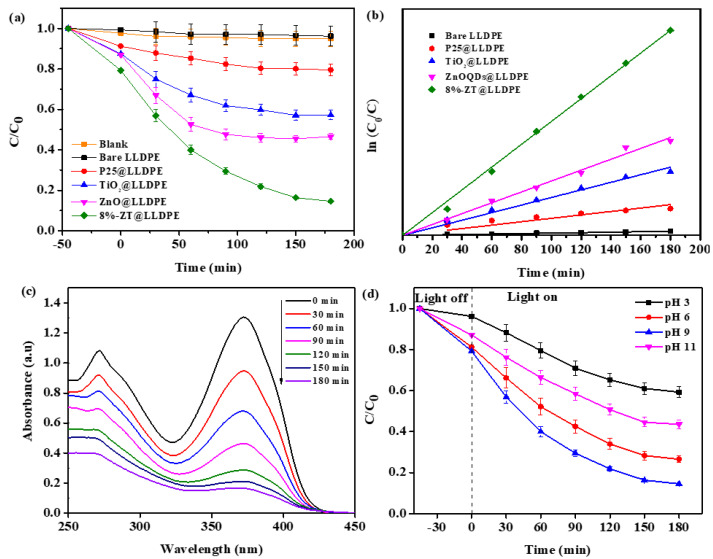
(**a**) Photocatalytic degradation of TC with various LLDPE hybrid films ([TC] = 40 mg/L; pH 9; lamp power: 48 W); (**b**) the rate curve based on the Langmuir–Hinshelwood kinetic model; (**c**) UV spectral changes during photocatalytic degradation of TC using 8%-ZT@LLDPE composite ([TC] = 40 mg/L; pH 9; lamp power: 48 W); and (**d**) effect of initial pH of the solution ([TC] = 40 mg/L; lamp power: 48 W).

**Figure 8 molecules-26-02509-f008:**
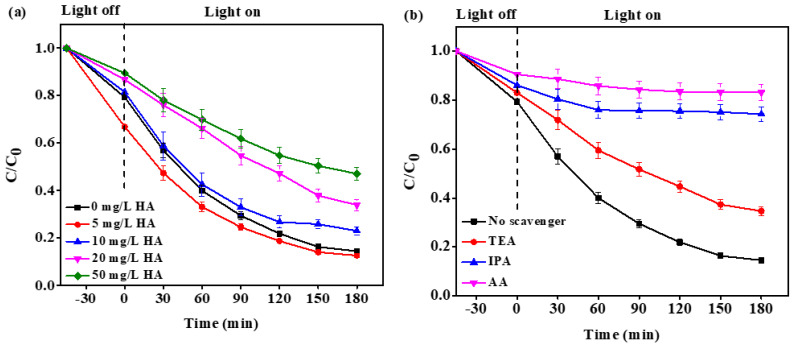
(**a**) Effect of HA on the TC removal; (**b**) the effect of radical scavengers on the photodegradation of TC using 8%-ZT@LLDPE film ([TC] = 40 mg/L; pH 9; lamp power: 48 W).

**Figure 9 molecules-26-02509-f009:**
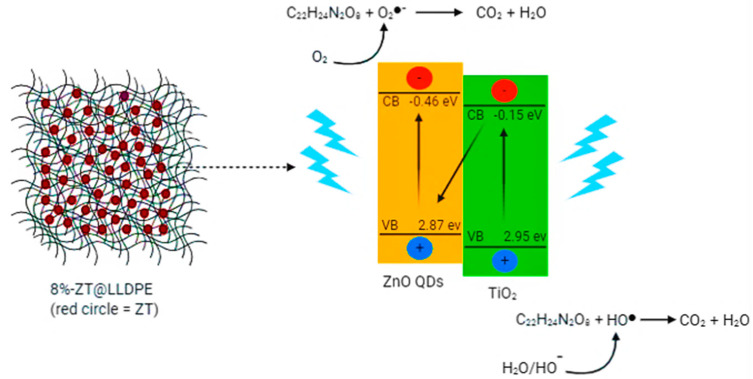
Proposed mechanism for the photocatalytic degradation of TC using 8%-ZT@LLDPE under visible light irradiation.

**Figure 10 molecules-26-02509-f010:**
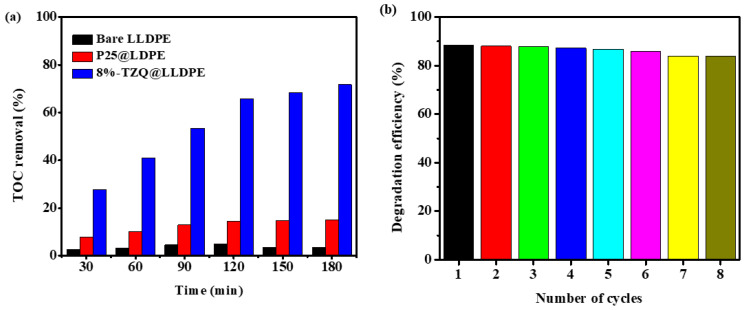
(**a**) Photocatalytic mineralisation of TC in the presence of bare LLDPE and LLDPE hybrid films, and (**b**) recycle experiment for the photodegradation of TC using 8%-ZT@LLDPE.

**Table 1 molecules-26-02509-t001:** Kinetic data for the photodegradation of TC using LLDPE hybrid films.

Catalyst	Efficiency (%)	k (min^−1^)	*R* ^2^
Bare LLDPE	4	0.0002	0.99724
P25@LLDPE	20.5	0.00151	0.96392
TiO_2_@LLDPE	42.7	0.00334	0.99503
ZnO@LLDPE	53.4	0.00480	0.99690
8%-ZT@LLDPE	85.4	0.01077	0.99897

**Table 2 molecules-26-02509-t002:** Comparison of recently reported studies on immobilised photocatalysts with the present study on the degradation of TC.

Photocatalyst	TC Conc.	Light Source	Results	Ref.
Au/Ag/TiO_2_ (cellulose acetate)	5 mg/L	Xe lamp (λ > 420 nm)	77.4% after 120 min (k = 0.01276 min^−1^).	[44]
TiO_2_ (alumina pellets)	35 mg/L	UV light	60% after 120 min (k = 0.0085 min^−1^)	[45]
TiO_2_ (chitosan)	40 mg/L	UV light	44% after 60 min (k = 0.0009 min^−1^) Reusability = 4 cycles	[46]
H_4_SiW_12_O_40_ (cellulose acetate)	10 mg/L	300W Hg lamp	63.8% after 120 min Reusability = 3 cycles	[47]
MoS_2_/BiOBr (carbon fibres)	20 mg/L	300 W Xe lamp	92.4% after 120 min Reusability = 4 cycles	[48]
Bi_2_WO_6_/ZnFe_2_O_4_ (polyurethane)	50 mg/L	500 W Xe lamp	91.59% after 90 min Reusability = 4 cycles	[49]
g-C_3_N_4_/rGO (3D Ni foam)	20 mg/L	300W Xe lamp	90% after 120 min Reusability = 4 cycles	[50]
8%-ZT@LLDPE	40 mg/L	48W fluorescent lamp	89.45% after 180 min (k = 0.01312 min^−1^) Reusability = 8 cycles	Present study

## Data Availability

Not applicable.

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
