# Peer review of "Floating ZnO QDs-Modified TiO_2_/LLDPE Hybrid Polymer Film for the Effective Photodegradation of Tetracycline under Fluorescent Light Irradiation: Synthesis and Characterisation"

_molecules, 2021, doi:10.3390/molecules26092509_

Round 1

Reviewer 1 Report

Dear Authors

97-110 ZnO should give diffraction even nanometer scale. You should do XRD for ZnO QDs.

97-110 How could you deduced it's size?

97-110 The explanation of strain or stress is not adequate. The same holds for the explanation about the incorporation of ZnOQDs in TiO2.

You should do TEM.

137 How did you observe nanoparticle agglomeration?

141 How did that "round" pores occured? You should do cross-section.

233 How does a neutral TC molecule experience electrostatic repulsion?

Author Response

Response to Reviewer Comments on MS ID:  molecules-1165168

Reviewer #1

Comments /Suggestions

97-110 ZnO should give diffraction even nanometer scale. You should do XRD for ZnO QDs.

Response

Thank you for the suggestion. The XRD of ZnO QDs has been added in Figure 1. The explanation has been included in Section 2.1.1.

Comments /Suggestions

How could you deduced it's size?

Response

Thank you for the question. The crystallite size was determined using the Scherrer equation. The equation has been inserted in Section 2.1.1 as Equation 1.

Comments /Suggestions

97-110 The explanation of strain or stress is not adequate. The same holds for the explanation about the incorporation of ZnO QDs in TiO2.

Response

Thank you for the comment. The ZT synthesis was carried out in a one-pot synthesis technique in which the ZnO QDs was added to the TTIP solution. Hence, the framework of anatase TiO2 was built surrounding the ZnO QDs. The changes in the pH and high calcination may have induced some changes to the crystal structure of the ZnO QDs. The intensity of the XRD peaks related to anatase TiO2 reduced when incorporated with LLDPE, suggesting the change in the crystal structure of the anatase TiO2. Thus, it is possible similar structural changes take place to the ZnO QDs crystal structure. We have restructured the sentences to make the explanation clearer. We have also included other possibilities. Please refer to Section 2.1.1

More information has been added to explain the incorporation of ZnO QDs in TiO2. Please refer to  Section 2.1.2. Corrected sections are denoted in red highlights in the revised manuscript.

Comments /Suggestions

You should do TEM

Response

Thank you for the suggestion. The TEM analysis has been added Section 2.1.3. However, we are unable to do the analysis for 8%-ZT@LLDPE inability to access the instrument due to the very long queue. Even if we send the sample out external will likely take a longer time to obtain the results. We have done the cross-section of the 8%-ZT@LLDPE for SEM analysis. Refer section 2.1.2.

Comments /Suggestions

How did you observe nanoparticle agglomeration?

Response

Thank you for the question. The observation was done through SEM analysis. Please refer to Section 2.1.2. The bright spots found in the SEM images of ZnO containing samples is often associated with ZnO nanoparticles. To avoid confusion, we have removed the word agglomeration and substitute it with nanoparticles.  We have included a relevant reference as well.

Comments /Suggestions

How did that "round" pores occured? You should do cross-section.

Response

Thank you for the question. We have added an explanation of the pore formation. Please refer to Section 2.1.2. We have included the cross-section in Section 2.1.2

Comments /Suggestions

How does a neutral TC molecule experience electrostatic repulsion?

Response

Thank you for the question. We have made a mistake; the correct word should be attraction. The mistake has been corrected. The correct sentence  should be “At pH 6, less electrostatic attraction was experienced by the neutral TC molecule and the positively charged surface of the catalyst.”

In summary, the authors acknowledge Reviewer #1 for the constructive and helpful critical comments on this manuscript. We have reflected the responses in the revised manuscript along with extensive editing of the language, clarity, and syntax throughout.

Reviewer 2 Report

please, see the attached file

Author Response

Response to Reviewer Comments on MS ID:  molecules-1165168

Reviewer #2

Comments /Suggestions

Major revisions:-Considering the high importance of the optical properties of the synthesized photocatalyst, I suggest adding to the manuscript its optical characterization in terms of absorption spectra of pure TiO2, ZnO-QDs; LLDPEand the final 8%-ZT@LLDPE.

Response

Thank you for the suggestion. The optical properties have been added. Please refer to Section 2.1.6.

Comments /Suggestions

-In figure 5a, a comparison between the activity of different substrates is performed, both in dark (simple adsorption) and under illumination. Even if the comparison of the here-synthesized 8%-ZT@LLDPE with an analogous composite system obtained by employing commercial TiO2 photocatalyst (P25) can be useful, I think that this graph could be improved by adding the data related to TiO2@LLDPE, using the here-synthesized TiO2, in order to better clarify the role of the different components in the final system.Also,the test ofa system obtained by combining ZnO QDots and LLDPE couldbe useful, in this regard.

Response

Thank you for the suggestion. We have included the graphs in Section 2.1.1.

Comments /Suggestions

Minor revisions:-As authors observe in the introduction, TiO2is one of the most studied and employed photocatalyst, but one of the main drawbacks of its use, especiallyin the form of nano-powder, is that its recovery after use is difficult. As authors suggest, combining it with polymeric materials is a viable solution to overcome this limitation. In this regard, I suggest citing the following paper, in which a similar strategy has been followed, by combining TiO2NPs whit alginate matrix. I. Vassalini, J. Gjipalaji, S. Crespi, A. Gianoncelli, M.Mella, M. Ferroni, I. Alessandri, “Alginate-Derived Active Blend Enhances Adsorption and Photocatalytic Removal of Organic Pollutants in Water”, Adv. Sustainable Syst. 2020, 1900112.

Embedding active photocatalyst into polymeric matrix, in order to improve the environmental sustainability of the decontaminant and reduce environmental concerns,is a strategy that can be extended also to other typologies of photocatalyst, as such as plasmonic AgNPs.

 In this regard, I suggest the following reference I. Vassalini, G. Ribaudo, A. Gianoncelli, M.F. Casula, I. Alessandri, “Plasmonic hydrogels for capture, detection and removal of organic pollutants”, Environ. Sci.: Nano, 2020,7, 3888-3900-

Response

Thank you for the suggestion. The references have been added as [18] and [19]

Comments /Suggestions

I think that, in the present version of the manuscript, the definition of the acronym ZT is missing. For example, it should be defined at line 85. “In this work, floating ZnO QDs modified TiO2(ZT)was supported on LLDPE hybrid film was synthesized through a facile and eco-friendly solution casting method.”-

Response

Thank you for the comment. ZT is used as a label and does not carry any definition. We have removed ZT from the sentence. We have included ZT as the label of the nanocomposite under methodology. Please refer to 3.3 line 532.

Comments /Suggestions

Regarding XRD characterization, at lines 101-102 the authors state that “The diffraction peaks related to the ZnO QDs were not observed”, due to their small size. The same is restated at lines 108-109. Anyway, at line 103, they say “The size of the ZnO QDsweredetermined using the Scherrer Equation was 4.4 nm.”. It is not clear to me how it has been possible to use the Scherrer equation in absence of any XRD peak.

Response

Thank you for the comment. We have included the XRD diffractogram for ZnO QDs. Please refer to Section 3.3.

Comments /Suggestions

-In figure 3, something is missing. Maybe AFM image of ZT?

Response

AFM was not done to ZT because our aim was to investigate the changes of the LLDPE film in terms of roughness. In our opinion, SEM analysis provides adequate information on the morphology of ZT. We have also included TEM images of ZT. We have removed (a) from the figure caption. Please refer Section 2.1.4.

Comments /Suggestions

I suggest adding, maybe in the SI, the emission spectrum of the lamps used during the photodegradation experiments.-

Response

We do not have the facility used to measure the emission spectrum of the lamps used in School of Chemical Sciences,Universiti Sains Malaysia. However, we have measured the light intensity and residual UV leakage. Please refer to Section 3.6. We believe it is adequate to include the light intensity and residual UV leakage.

Comments /Suggestions

-Please, specify the pH used for achieving graph in Figure 5a.

Response

Thank you for the comment. pH 9 was used and it is included in the figure caption of Figure 7(a). Figure 7. (a) Photocatalytic degradation of TC using various LLDPE hybrid films ([TC] = 40 mg/L; pH 9; lamp power: 48 W).

Comments /Suggestions

-I suggest controlling the figure numbers along the text. I think that it should be Figure 7 at line 276, Figure 8 at line 308 and Figure 9 at line 336.

Response

Thank you for the comment. We have corrected all the Figures.

Comments /Suggestions

-About the mechanism of TC photodegradation, in figure 8 a possible pathway is reported,and mass spectra are reported in Figure S4. Are they related to different samples collected after different irradiation time? Please, add this information.

Response

Thank you for the comments. The information has been added. Refer line 432 and 433. It was also mentioned in line 591.

In summary, the authors acknowledge Reviewer #2 for the constructive and helpful critical comments on this manuscript. We have reflected the responses in the revised manuscript along with extensive editing of the language, clarity, and syntax throughout.

Round 2

Reviewer 1 Report

I have no comments

Reviewer 2 Report

The authors answered all my comments and I suggest accepting the manuscript in the current form.

Author Response

We would like to thank the reviewer for his valuables commments and we appreciate his time and effort in reviewing our manuscript.